

# Measurement Report: Effects of transition metal ions on the optical properties of humic-like substances revealing structural preference

Juanjuan Qin[1,2], Leiming Zhang[3], Yuanyuan Qin[1], Shaoxuan Shi[1], Jingnan Li[1], Zhao Shu[1], Yuwei Gao[1], Ting Qi[4], Jihua Tan[1], Xinming Wang[2]

[1]College of Resources and Environment, University of Chinese Academy of Sciences, Beijing 100049, China
[2]Guangzhou Institute of Geochemistry, Chinese Academy of Sciences, Guangzhou 510640, China
[3]Air Quality Research Division, Science & Technology Branch, Environment and Climate Change Canada, Toronto, Canada
[4]College of Resources and Environment, University of Chinese Academy of Sciences, Beijing 100049, China
*Correspondence to*: Jihua Tan (tanjh@ucas.ac.cn); Xinming Wang (wangxm@gig.ac.cn)

**Abstract.** Humic-like substances (HULIS) are complex macromolecules in water-soluble organic compounds containing multiple functional groups, and transition metal ions (TMs) are ubiquitous in atmospheric particles. In this study, potential physical and chemical interactions between HULIS and four TM species including $Cu^{2+}$, $Mn^{2+}$, $Ni^{2+}$, and $Zn^{2+}$ were analyzed by optical method under acidic, weakly acidic and neutral conditions. The results showed that $Cu^{2+}$, $Mn^{2+}$, and $Zn^{2+}$ only slightly enhanced mass absorption efficiency ($MAE_{365}$) of HULIS in winter, and had indiscernible effects on absorption Ångström exponent (AAE) of HULIS in both seasons under all acidity conditions. All four TMs had fluorescence quenching effects on winter HULIS, and only $Cu^{2+}$ had similar effects on summer HULIS, with the highest quenching coefficients found under weakly acidic condition in both seasons. The $^1$H-nuclear magnetic resonance ($^1$H-NMR) and Fourier-transform infrared (FT-IR) spectra revealed that $Cu^{2+}$ mainly bound with aromatic species and tightened the molecule structures of HULIS. The parallel factor analysis (PARAFAC) results extracted four components of HULIS, including low-oxidized humic-like substances (C1), N-containing compounds (C2), highly-oxidized humic-like substances (C3), and the mixing residentials (C4), from the fluorescence spectra in both winter and summer. The divergent variations of HULIS spectral components with $Cu^{2+}$ additions under three acidity conditions indicated that electron-donating groups of HULIS mainly corresponded to C1 and C3, with $Cu^{2+}$ binding with HULIS by replacing proton, while electron-withdrawing groups of HULIS could correspond to C2, with its connection with $Cu^{2+}$ through electrostatic adsorption or colliding induced energy transfer.





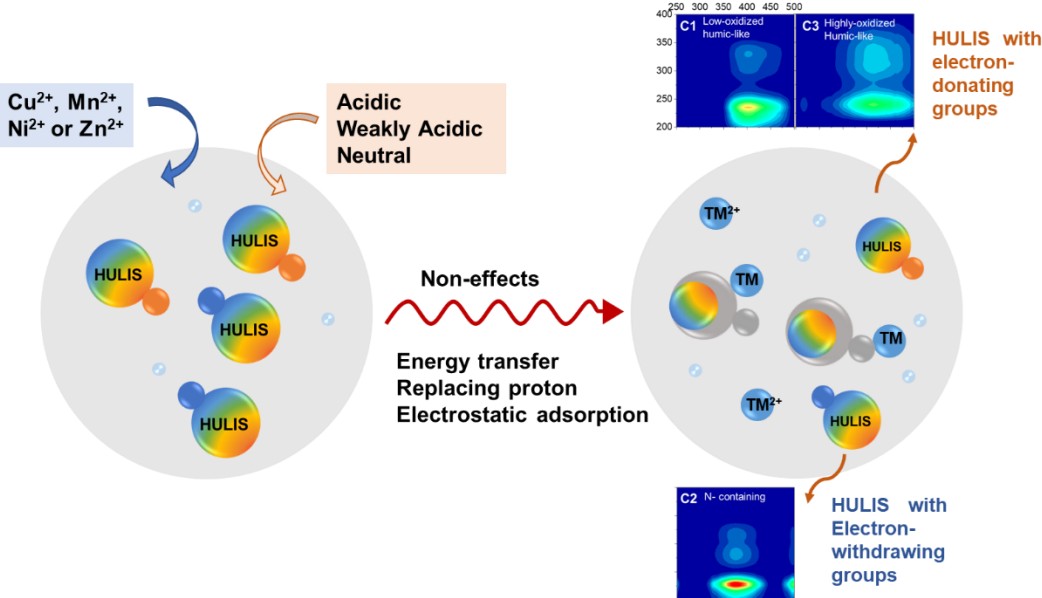

## 1 Introduction

Atmospheric particles, consisting of both organic and inorganic substances, have significant impacts on the natural environment, climate, and human health. Humic-like substances (HULIS) and transition metal ions (TMs) are among the

major organic and inorganic fractions, and can substantially affect the physical and chemical properties of atmospheric particles (Hawkins et al., 2016; Zanca et al., 2017; Frka et al., 2018; Ma et al., 2019). Besides, HULIS and many TMs, like Fe, Cu, Mn, V, Cr, and Co, have negative impacts on human health since they can promote the generation of reactive oxygen species (ROS), which cause inflammatory response of human respiratory system (Verma et al., 2012; Gali et al., 2015; Lin and Yu 2019). HULIS are known as a mixture of macromolecular organic compounds, containing aromatics and aliphatic

species with multiple oxygenated functional groups like carbonyls, hydroxyl, nitrate, and nitroxy-organosulfate (Win et al., 2018). TMs can transfer electrons and participate in chemical reactions or serve as catalyst, especially in atmospheric photochemistry (Mao et al., 2013; Guasco et al., 2014).

Considering that HULIS and TMs are both active chemical components of atmospheric particles, the significance of their co-effects is apparently conceivable. Early studies have demonstrated that HULIS can enhance the solubility of TMs by organic

complexation (Paris and Desboeufs 2013; Scheinhardt et al., 2013), and TMs might mediate organic transformation processes and induce the formation of light absorbing secondary water-soluble organics and water insoluble abiotic polymers (Slikboer et al., 2015). More recent studies focused on the oxidation potentials, and ROS formation or other metal induced photochemical reactions of TMs ($Fe^{2+}$, $Cu^{2+}$, and $Mn^{2+}$) in combination with HULIS (Gonzalez et al., 2017; Lin and Yu 2019; 2020; Ye et al., 2021). The mixture of TMs and HULIS could exhibit synergistic or antagonistic effects on ROS formation



associated with the chemical composition of HULIS. Moreover, TMs can also bind with HULIS and affect the light
absorption properties of HULIS (Fan et al., 2021; Wang et al., 2021).

To reveal the linkage between HULIS and TMs in atmospheric chemistry, this study characterized the varying optical and
chemical properties of HULIS with addition of different TMs species under three different levels of acidity. The effects of
TMs on light absorption of HULIS and the TMs-sensitive fraction of HULIS were determined by optical spectra, and

parallel factor analysis (PARAFAC) were performed to unveil the susceptive fluorescent components. The interaction
between $Cu^{2+}$ and HULIS were further analyzed for its prominent effects, and the potential reaction mechanisms were
deduced based on the structural and optical variation scenarios. Results from the present study advanced our understanding
on the physical and chemical characteristics and transformations of HULIS, especially the physical and chemical
connections between HULIS and TMs.

## 2 Methods

### 2.1 Sample extraction and titration

In the present research, two concentrated HULIS solutions were extracted from $PM_{2.5}$ samples collected in a winter and
summer month in Beijing, and the detailed sample information can be found elsewhere (Qin et al., 2022). The HULIS
isolation was mainly based on the method established by Lin et al., (2010) with some adjusted procedures for practical

reasons. Firstly, WSOC was ultrasonically extracted from 30 or 60 daily $PM_{2.5}$ filter samples (D = 2 cm, 30 pieces for winter
and 60 pieces for summer), using 200 mL Milli-Q water three times, and filtered using 0.22 μm membranes. The liquids
were then separated by SPE cartridges (Oasis HLB, 6cc, 500 mg/cartridge, Waters, USA), rinsed with 10 mL water, and
pumped to dry. The cartridges were eluted with 10 mL methanol afterwards. Finally, the eluants were evaporated to dryness
by $N_2$ at room temperature and redissolved into 200 mL ultrapure water to obtain a HULIS stock solution.

Before metal ion titration, the HULIS stock solutions were diluted by 10 folds, and the total organic carbon (TOC)
concentration was determined by a TOC analyzer (Multi N/C 3100, Germany), with the carbon concentrations of 1.9 mg/L
for winter HULIS and 1.6 mg/L for summer HULIS. The original acidity of HULIS was determined by a pH meter (Mettler
Toledo, Swiss) before and after dilution. In both winter and summer, triplicate 600 mL of diluted HULIS solutions were
separately added to 1 L sealed vials to obtain weakly acidic HULIS (pH = 5.65 for winter and pH = 5.92 for summer), acidic

HULIS (pH ≈ 3.0, by 0.05M $H_2SO_4$), and neutral (pH ≈ 7.5, by 0.1M NaOH), respectively. Considering the instability of
natural pH gradient, excessive base solution was added to ensure the solution pH to be around neutral before measurement.
Four transition metals ions including $Cu^{2+}$, $Zn^{2+}$, $Mn^{2+}$, and $Ni^{2+}$ were chosen as representative metal ions that might be
reactive to HULIS, considering their richness in atmospheric environment or significant health risks (Fan et al., 2021; Wang
et al., 2021). Titration experiments were conducted by adding 0 – 526.32 μl corresponding metal sulfate solutions (0.01 M)

and 9 ml diluted HULIS solution to a 10 ml amber volumetric flask, and ultrapure water was then added to obtain samples
containing TMs concentrations of 0, 20, 50, 100, 200, 300 and 500 μM, respectively. The titrated solutions were shaken for



12 h at room temperature to ensure complexation equilibrium, and pH of all titrated samples were detected again before spectral analysis.

## 2.2 Spectral analysis

The optical properties of HULIS with addition of TMs (0–500 μM) were analyzed by UV-Vis and fluorescence spectrophotometers under acidic, neutral, and neutral pH. The chemical properties of HULIS before and after adding TMs were analyzed by [1]H-nuclear magnetic resonance ([1]H-NMR) and Fourier-transform infrared (FT-IR).

The UV-vis absorption spectra of WSOC and HULIS were measured by UV-VIS photometer in Agilent Technologies covering wavelength range of 200 nm–500 nm with an interval of 1nm, using a scanning speed of 200 nm/min. The sample

pool is a quartz colorimetric dish with an optical path of 1 cm.

The three-dimensional fluorescence spectra were measured by an Agilent Technologies photometer under the following conditions: excitation wavelength (EX) 200 nm–400 nm, step length 5 nm, slit width 10 nm; emission wavelength (EM) 250 nm–500 nm, step length 5 nm, slit width 5 nm; scanning speed: 2400 nm/min; and response time: automatic. The sample pool uses a quartz colorimeter with a light path of 1cm.

The infrared spectra of HULIS for wavenumber between 4000-400 cm$^{-1}$ were measured by an FTIR spectrometer (Thermal Fisher, Nicolet IS10) at room temperature. The essential extraction of HULIS for FT-IR measurements was the same as those described in section 2.1 above, except that around 1 mg of the HULIS (dried by gentle stream of $N_2$ at room temperature 25 ℃) was directly mixed with 60 mg KBr and dried in a drying oven, rather than redissolved by water. The mixtures were then ground and pressed into firm tablets for analysis. A 600 MHz NMR spectrometer (Bruker, AVANCE III

600 M) was applied for [1]H-NMR spectra measurements. About 5 mg HULIS samples were redissolved into 1 mg $D_2O$ for detection.

## 2.3 Data Analysis

### 2.3.1 Mass absorption efficiency and absorption Ångström exponent

The mass absorption efficiency (MAE, $10^3 \cdot AU \cdot cm^2/mg$) of HULIS and the absorption Ångström exponent (AAE) were

obtained as follows (Kirillova et al., 2014; Saleh et al., 2014):

$$MAE = (A_\lambda - A_{700}) \times \frac{\ln(10)}{C_{spcies} \times L} \tag{1}$$

$$A_\lambda = K\lambda^{-AAE} \tag{2}$$

Where $A_\lambda$ is light absorbance at wavelength λ, $C_{species}$ is the chemical concentration of organic compounds (WSOC and HULIS in the present research), L is the light path length (1 cm), $K$ is a scaling constant, and the fitting wavelength of AAE

is 330-400 nm.



### 2.3.2 Fluorescence indices

Fluorescence indices based on intensity ratios may provide clues about the condensation state of WSOC. The average fluorescence intensity is the mean intensity of an excitation-emission matrix (EEM) spectra; the specific fluorescence intensity (SFI) is the fluorescence intensity per unit TOC (mg/L) (Kalbitz et al., 2000). The biological index (BIX) reflects

the freshness of microbially produced DOM (Parlanti et al., 2000), with a high BIX value (> 1) corresponding to a predominantly microbial derived organic matter, and a value of <0.6 indicating fewer microbial originated DOM (Qin et al., 2018). BIX is calculated as follows:

$$BIX = \frac{EEM_{Ex_{200-310},Em_{250-380}}}{EEM_{(max\,(Ex_{200-310},Em_{420-435}))}} \quad (3)$$

The fluorescence lifetime for HULIS was estimated by a formula derived from the Strickler–Berg equation (Strickler and

Berg 1962). The $\tau_0/\rho$ ratio comprehensively reflects the fluorescence intrinsic lifetime (related to the fluorophore type) and fluorophore density (associated with the molecular structure), a large $\tau_0/\rho$ can be resulted from a long fluorescence lifetime or a small fluorophore (Xiao et al., 2018), it was a representation of fluorescence lifetime in this research.

$$\tau_0/\rho = c2.88 \times 10^{-12} n^2 \frac{\int_{Em} FI(v)dv}{\int_{Em} v^{-3} FI(v)dv} \int_{Ex} UVA(v)dv^{-1} \quad (4)$$

Where $\tau_0$ is the intrinsic lifetime (ns), $\rho$ is the density of fluorophore in the organics (mmol-fluorophore per gC), $c$ is the total

organic concentration (mgC/L), $n$ is the refractive index of the solution ($n^2 \approx 1.8$), $v$ is the excitation or emission wavenumber (cm$^{-1}$), UV is the UV-Vis absorbance (A.U.), and $FI$ is the fluorescence intensity (R.U.).

Stokes shift (SS) can reflect the fluorescence energy of organics (Xiao et al., 2020; Yu et al., 2020), and is defined as Ex$^{-1}$-Em$^{-1}$ nm$^{-1}$, demonstrating the energy loss of the excited fluorophore due to internal conversion during relaxation (Lakowicz 2006). AvgSS were defined as the intensity-weighted average Stokes shifts of HULIS (Xiao et al., 2019).

### 2.3.3 Parallel factor analysis (PARAFAC)

PARAFAC model can decompose complex EEM spectra into several main components by statistical method. The excitation spectrum, emission spectrum, and scores of each component are as follows:

$$x_{ijk} = \sum_{f=1}^{F} a_{if} b_{jf} c_{kf} + \varepsilon_{ijk} \quad i=1,\dots,I\ j=1,\dots,J\ k=1,\dots,K \quad (5)$$

Where $x$ represents the fluorescence intensity, $f$ is the number of components resolved by PARAFAC, $a$ is proportional to the

concentration of the $f$-th component, and $b$ and $c$ are the scaled estimating of the emission and excitation spectra. The subscript $i$ is the sample number, and $j$ and $k$ represent emission and excitation wavelength, respectively. Before performing PARAFAC, all EEM data were normalized to unit norm to reduce concentration-related collinearity and avoid extremely different leverages (Wang et al., 2020). Tucker congruence coefficient (TCC) was determined for each excitation spectrum and emission spectrum, and a threshold of 0.95 was applied to confirm the spectral congruence. The model was determined

by half-split validation.



## 3 Results and discussion

The light absorption and fluorescence efficiencies of winter and summer HULIS under acidic, weakly acidic, and neutral environments are depicted in Figure 1, and the spectra derived indices are listed in

Table 1. Generally, the overall light absorption efficiencies (represented by MAE) of winter HULIS were higher than those in summer, and the AAE showed a reverse trend, indicating higher light absorption abilities of winter than summer samples (Mo et al., 2021). The MAE and AAE of HULIS under three acidity levels in both seasons were consistent with those reported in our previous research, with the average $MAE_{365}$ at 0.011±0.00 in winter and 0.005±0.001 in summer, and the corresponding average $AAE_{300-400nm}$ at 6.46±0.86 and 6.97± 0.83, respectively (Qin et al., 2022). The fluorescence efficiencies of winter HULIS (represented by SFI and AFI/TOC) were higher than those of summer HULIS as well (Figure 1), and the fluorescence indices reveled some structural differences. The SFI spectra of HULIS mainly exhibited two fluorescence peaks at wavelength of Ex = 225–230 nm, Em = 380–400 nm, and Ex = 325–330 nm, Em = 395–400 nm in both seasons, with the peaks being able to be roughly characterized as fulvic-acid like group and protein-like groups, respectively. The Stokes shift index of AvgSS could be associated with the hydrophobicity/hydrophilicity of organics, with a low AvgSS implying a relatively high hydrophilicity and a high one implying a high hydrophilicity (aromaticity or humification degree) (Xiao et al., 2019). Thus, the higher AvgSS of winter HULIS indicated a higher hydrophilicity compared to that in summer.

The different optical properties of HULIS between winter and summer were mainly resulted from divergent chemical structures. Our previous research had revealed that winter HULIS contained more unsaturated structures and were relatively hydrophobic, leading to the higher light absorption and fluorescence efficiencies, while summer HULIS were rich with oxidized species and functional groups and were more hydrophilic (Qin et al., 2022). Baduel et al., (2011) evaluated the varying optical properties of HULIS during $O_3$ oxidation, and found the C=O and COOH groups of HULIS increased with oxidation. However, Yang et al., (2021) found that $O_3$ oxidation of humic acid could consume some functional groups like aromatic C=C, OH, C=O, COOH and $COO^-$, causing an decrease in MAE and an increase in AAE, and these divergences were ascribed to the different structures of HULIS. The different structures of HULIS between winter and summer could not only lead to the variated optical properties but also the distinctive behaviors of HULIS with other coexist species in atmospheric environment. Therefore, it is necessary to analyze the relations between TMs and HULIS for winter and summer HULIS separately.



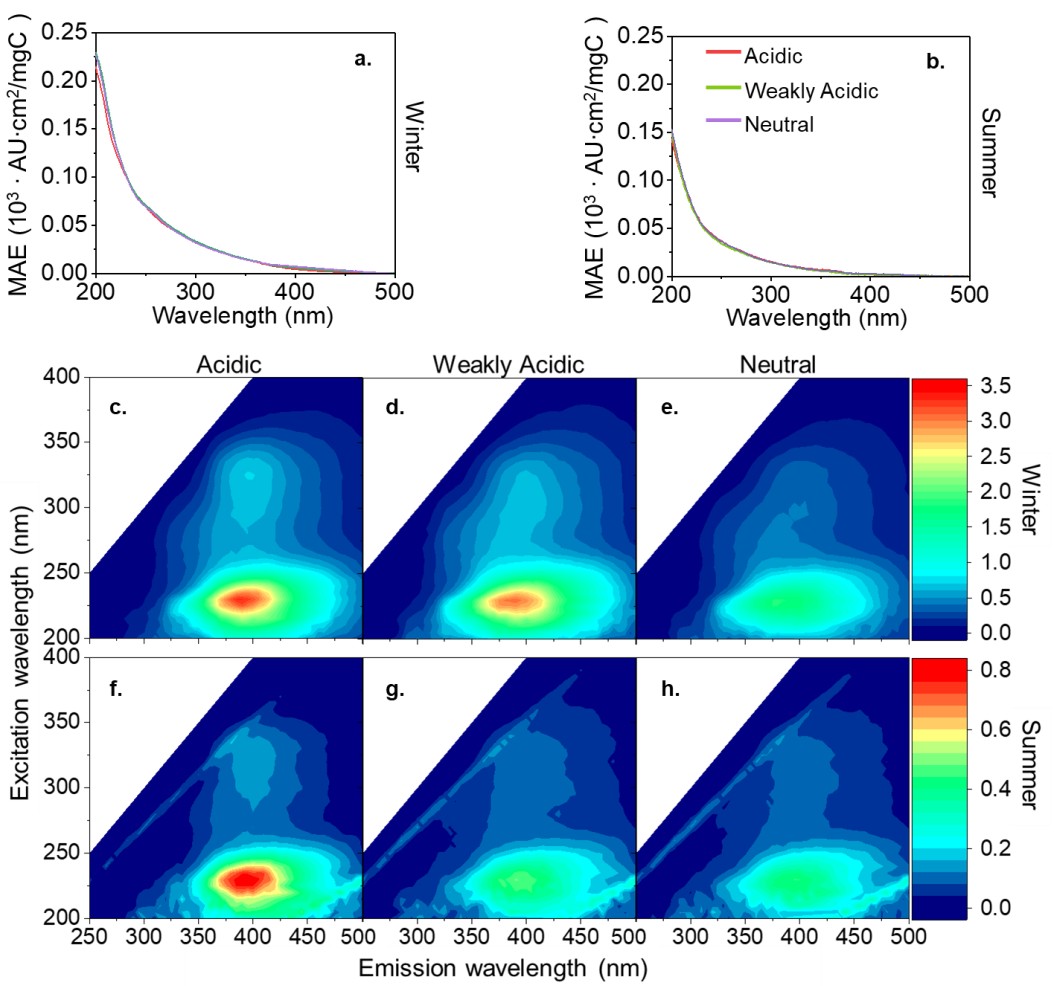

**Figure 1 The MAE (a-b) and SFI (c-h) spectra of winter and summer HULIS under three acidity conditions.**

**Table 1 The light absorption and fluorescence indices of HULIS under three acidity conditions.**

|  | Winter |  |  |  | Summer |  |  |  |
|---|---|---|---|---|---|---|---|---|
|  | Acidic | Weakly acidic | Neutral | Mean ± S.D. | Acidic | Weakly acidic | Neutral | Mean ± S.D. |
| $MAE_{365}$[a] | 0.011 | 0.011 | 0.011 | 0.011 ± 0.00 | 0.006 | 0.004 | 0.006 | 0.005 ± 0.001 |
| AAE | 7.05 | 6.86 | 5.47 | 6.46 ± 0.86 | 6.97 | 7.80 | 6.15 | 6.97± 0.83 |
| AFI/TOC[b] | 0.79 | 0.62 | 0.58 | 0.66 ± 0.11 | 0.10 | 0.10 | 0.09 | 0.10 ± 0.01 |
| BIX | 1.07 | 0.96 | 0.99 | 1.00 ± 0.05 | 0.98 | 0.90 | 0.90 | 0.93 ± 0.05 |





| | | | | | | | | |
|---|---|---|---|---|---|---|---|---|
| AvgSS | 1.54 | 1.54 | 1.54 | $1.54 \pm 0.00$ | 1.01 | 0.90 | 0.91 | $0.94 \pm 0.06$ |
| $\tau_0/\rho$ | 0.15 | 0.24 | 0.13 | $0.17 \pm 0.06$ | 0.56 | 0.92 | 0.60 | $0.69 \pm 0.19$ |

a: $10^3 \cdot AU \cdot cm^2/mgC$   b. $RU \cdot L/mgC$

### 3.2 Effects of TMs on the light absorption properties of HULIS

The gradients of $MAE_{365\_i}/MAE_{365\_0}$ and $AAE_{300-400nm}$ for HULIS with TM ($Cu^{2+}$, $Mn^{2+}$, $Ni^{2+}$, and $Zn^{2+}$) concentrations ranging from 0 to 500 μM under three acidity conditions are depicted in Figure 2 and the whole MAE spectra are exhibited
in SI Figures S1 and S2. For winter HULIS, under acidic environment, the addition of $Mn^{2+}$, $Ni^{2+}$, and $Zn^{2+}$ at low concentrations (< 50 μM) could induce a decrease of 14%–16% in $MAE_{365}$, and further increasing TM concentrations had little effects on $MAE_{365}$, whereas $Cu^{2+}$ showed minimal effects on $MAE_{365}$, which only decreased $MAE_{365}$ by 5% with increasing TM concentrations. Under weakly acidic environment, the addition of a small amount of any of the four TMs (0– 50 μM) could induce evident increase of $MAE_{365}$ at a range of 16% to 20%, and then $MAE_{365}$ became steady or slightly
decreased with further increasing TM concentrations. As for the neutral environment, the addition of $Cu^{2+}$ could lead to up to 9% increase in $MAE_{365}$, but $Cu^{2+}$ showed inhibiting effects on $MAE_{365}$ in cases with additions of $Mn^{2+}$, $Ni^{2+}$, or $Zn^{2+}$, with the maximum decrease at 11%, 7%, and 14%, respectively. Nevertheless, TMs could only change the $AAE_{300-400nm}$ of HULIS slightly, and the solution acidity sieved the initial $AAE_{300-400nm}$ of HULIS; besides, the TM concentrations had inconspicuous effects on the regularity. An exception was observed for $Ni^{2+}$ because of its self-absorption, with $AAE_{300-400nm}$
continuously reducing with increasing additions of $[Ni^{2+}]$.

For summer HULIS, $Cu^{2+}$, $Mn^{2+}$, and $Zn^{2+}$ could cause fluctuations in $MAE_{365}$ with increasing metal ion concentrations under acidic and neutral environments, and they inhibited the light absorption efficiencies of HULIS under weakly acidic environment. Likewise, $Ni^{2+}$ could enhance the $MAE_{365}$ of HULIS under all acidity conditions because of the increasing absorption generated by increasing $Ni^{2+}$ concentrations. The $AAE_{300-400nm}$ of HULIS had subtle variation when adding $Cu^{2+}$,
$Mn^{2+}$, and $Zn^{2+}$ under acidic and neutral environments, indicating that the TMs-HULIS mixtures might not compose complexations or their complexes don't have structural effects on HULIS. Yet when the solution was weakly acidic, the $AAE_{300-400nm}$ exhibited sharp fluctuations with increasing $Mn^{2+}$ and $Zn^{2+}$, and even increased with increasing $Cu^{2+}$ concentrations.



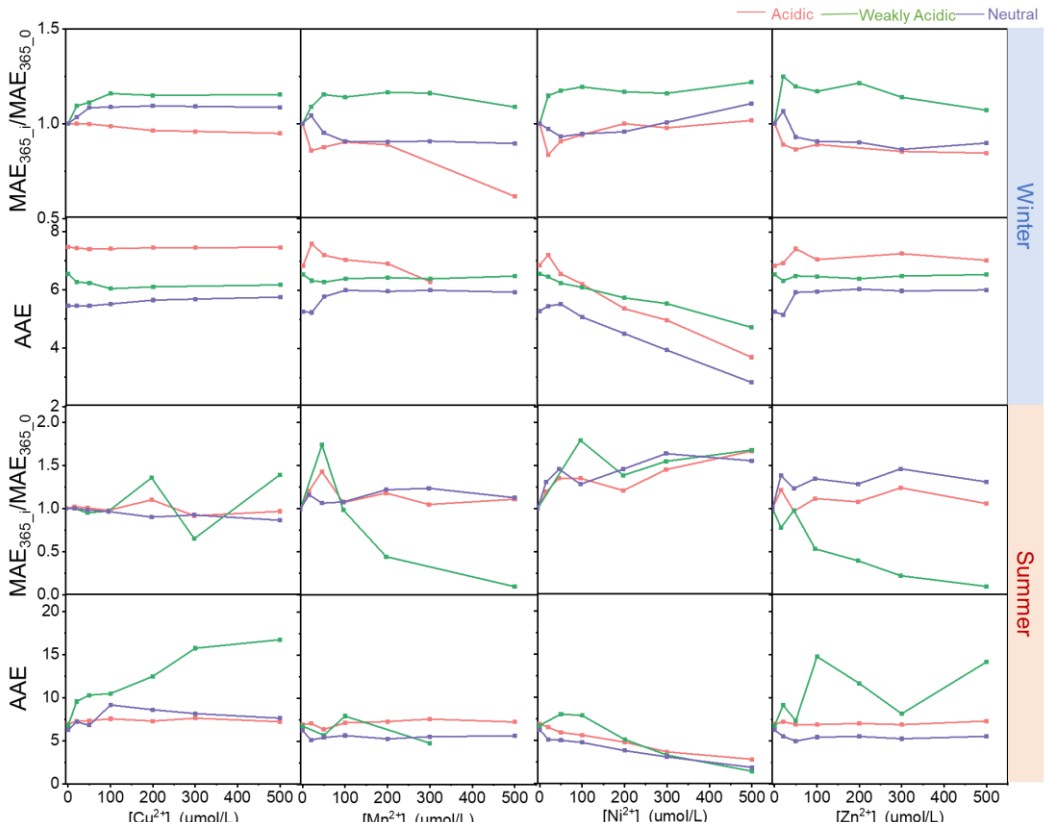

**Figure 2 The light absorption indices of HULIS with additions of TMs under different acidity conditions.**

## 3.3 Effects of TMs on the fluorescence properties of HULIS

The effects of TMs on the fluorescence indices of HULIS under different acidity environments are shown in Figure 3 and Figure 4, and the fluorescence spectral variations are provided in SI Figures S3 and S4. Obviously, all four TMs showed quenching effects on the AFI of HULIS with fiddling seasonal or acidity dependent differences, with the effectiveness ranking in the order of $Cu^{2+} > Mn^{2+} \sim Ni^{2+} > Zn^{2+}$. At the maximum TMs concentration of 500 μM, AFI of winter HULIS reduced by 5%–21%, 5%–11%, and 7%–28%, and of summer HULIS reduced by 8%–22%, 12%–39%, and 5%–26% under acidic, weakly acidic, and neutral environments, respectively.

The effects of TMs on fluorescent HULIS could be evaluated by the Stern-Volmer equation. Supposing metal ions only singly bonded with HULIS molecules, the quenching constants ($k$) of TMs species to the total fluorescence of HULIS were obtained and are listed in Table 2. Briefly, for winter HULIS, all TMs species had significant liner correlations with $AFI/AFI_0$ ($R^2 > 0.6$). $Cu^{2+}$ and $Mn^{2+}$ had largest $k$ constants under acidic and neutral environments and relatively low $k$ under weakly acidic environment, indicating that $Cu^{2+}$ and $Mn^{2+}$ might mainly bond with organic groups that dissociated under acidic or neutral environments. $Ni^{2+}$ had the same $k$ under weakly acidic and acidic environments and larger $k$ under neutral environment, indicating that $Ni^{2+}$ might prefer to bond with neutral HULIS. Similarly, $Zn^{2+}$ had larger $k$ under acidic



environment, indicating a preference of acidic environment for the complexation of $Zn^{2+}$ and HULIS. For summer HULIS, $Cu^{2+}$ still had significant and large $k$ under acidic and neutral environments and relatively low $k$ under neutral environment. $Mn^{2+}$ only had a relatively large $k$ under acidic environment and a low $k$ under neutral environment, but no statistically significant $k$ under weakly acidic environment. Besides, $AFI/AFI_0$ of HULIS with $Ni^{2+}$ and $Zn^{2+}$ also had non-significant $k$, indicating negligible interactions between fluorescent HULIS and these two ions.

Although TMs showed quenching effects on HULIS, the locations of fluorescence peaks rarely moved with increasing TM concentrations. Thus, the fluorescence spectra-based indices only show minimal variations under different acidity conditions with increasing metal ions concentrations. In Figure 4, the Stokes shift spectra of HULIS separated into groups under three acidity conditions for both seasons. Only addition of $Cu^{2+}$ could hierarchically distribute the spectra slightly. The other three TMs showed inconspicuous effects, indicating that they could not cause substantial structural variations on fluorescent HULIS, as can be more visually presented by the ratio of $AvgSS_0/AvgSS_i$ (Figure 5). Additionally, the addition of $Mn^{2+}$, $Ni^{2+}$, and $Zn^{2+}$ didn't change the $\tau_0/\rho$ of HULIS either, which might confirm that they did not bound with HULIS molecules since fluorescent lifetime is a sensitive indicator vulnerable to the micro-environment surrounding detected objects (Coble, 2014).

**Table 2 The fluorescence quenching coefficients of TMs and HULIS obtained from Stern-Volmer equation**

| Stern-Volmer | Acidity | $Cu^{2+}$ | | $Mn^{2+}$ | | $Ni^{2+}$ | | $Zn^{2+}$ | |
| --- | --- | --- | --- | --- | --- | --- | --- | --- | --- |
| | | $k \cdot 10^3$ | $R^2$ | $k \cdot 10^3$ | $R^2$ | $k \cdot 10^3$ | $R^2$ | $k \cdot 10^3$ | $R^2$ |
| Winter | Acidic | 0.60 | 0.99 | 0.10 | 0.98 | 0.10 | 0.95 | 0.20 | 0.98 |
| | Weakly Acidic | 0.10 | 0.86 | 0.08 | 0.91 | 0.10 | 0.95 | 0.07 | 0.83 |
| | Neutral | 0.60 | 1.00 | 0.10 | 0.68 | 0.20 | 0.90 | 0.07 | 0.58 |
| Summer | Acidic | 0. 60 | 0.99 | 0.20 | 0.38 | -0.02 | 0.79 | 0.01 | 0.02 |
| | Weakly Acidic | 1.00 | 0.82 | -0.10 | 0.02 | -0.04 | 0.05 | -0.20 | 0.13 |
| | Neutral | 0. 40 | 0.88 | 0.05 | 0.63 | 0.05 | 0.08 | -0.20 | 0.16 |

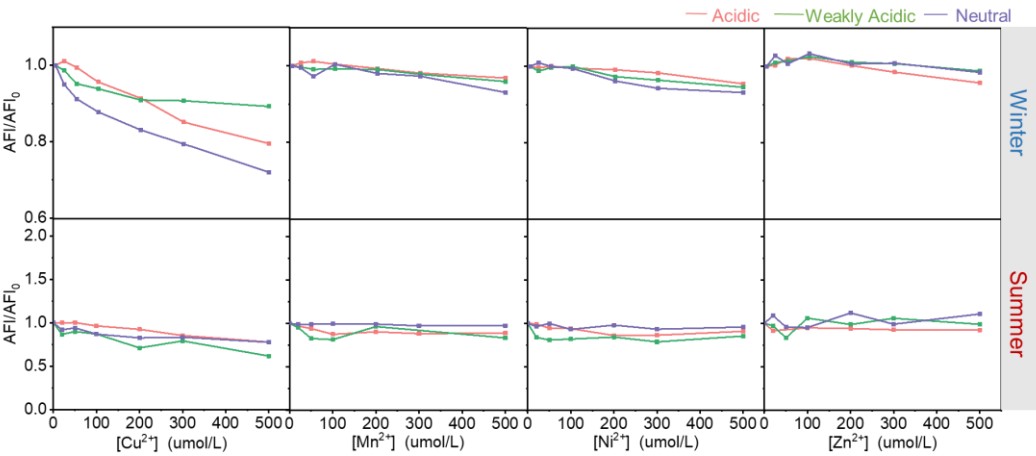

**Figure 3 The average fluorescence intensities of HULIS with adding of $Cu^{2+}$, $Zn^{2+}$, $Mn^{2+}$, and $Ni^{2+}$ under different acidity conditions in winter and summer.**





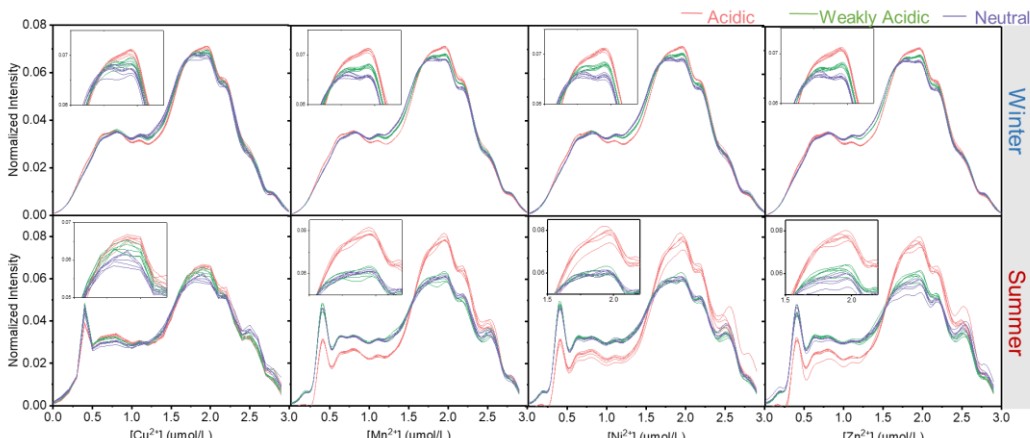

**Figure 4 The Stokes shift spectra of HULIS with adding of Cu²⁺, Zn²⁺, Mn²⁺, and Ni²⁺ under different acidity conditions.**

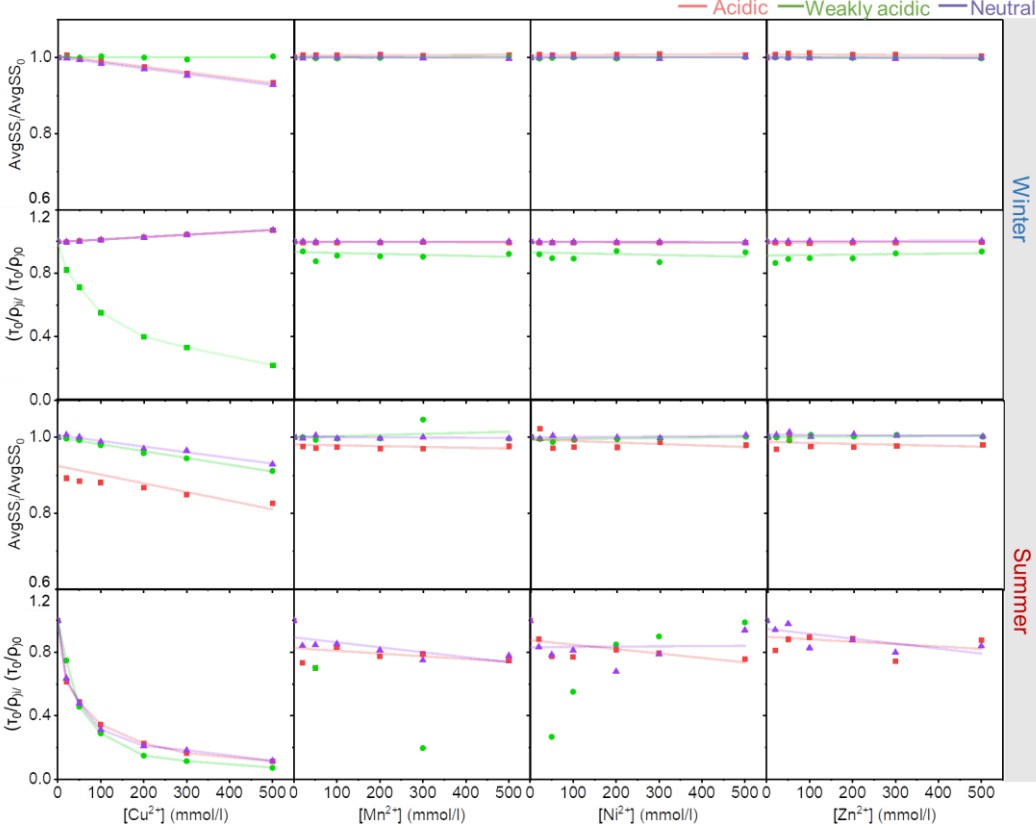

**Figure 5 The fluorescence indices of HULIS with increasing Cu²⁺, Zn²⁺, Mn²⁺, and Ni²⁺ concentrations under different acidity conditions.**





### 3.4 Distinguishing the varying fractions of HULIS with additions of TMs

To further identify the response of specific HULIS compounds to TMs, PARAFAC methods were applied to distinguish the varying groups of HULIS with addition of different TM species. In general, four compounds (C1-C4) were sorted out from

EEM spectra of winter and summer HULIS, and they were characterized as low-oxidized humic-like substances (C1), N-containing compounds (C2), highly-oxidized humic-like substances (C3), and the mixing residentials (C4), respectively (Chen et al., 2016). The constitution of C1-C4 in winter and summer HULIS under different acidity levels are listed in Table 3.

Form the correlations between $F/F_0$ ratio and TM concentrations of fluorophores C1-C4 shown in Figure 6 and Figure 7, it

was obvious that the intensities of four fluorophores had increasing or decreasing tendencies in response to the acidity and TM species because of the divergent acid/base groups of HULIS. In the case of winter HULIS under acidic environment, $Cu^{2+}$ could induce a substantial fluorescence quenching effects on C1 and C4 and a concurrent modest fluorescence increasing effects on C3, $Mn^{2+}$ and $Ni^{2+}$ exhibited weak fluorescence quenching effects on C1 and C3, and $Zn^{2+}$ exhibited weak fluorescence quenching effects on C1 and C4. Under weakly acidic environment, all of the four TMs exhibited

discernible quenching effects on C2 and C3, with $Cu^{2+}$ manifest as the most potent quencher. Under neutral environment, $Cu^{2+}$ significantly quenched the fluorescence intensities of C1, C2, and C4, $Ni^{2+}$ displayed modest quenching effects only on C3, and other metal ions showed inconspicuous fluorescence quenching/enhancement effects.

As for summer HULIS, only $Cu^{2+}$ exhibited notable quenching effects on fluorophores C1-C4, while $Mn^{2+}$, $Ni^{2+}$, and $Zn^{2+}$ showed rather minimal or erratic effects under all acidity conditions. Under acidic and weakly acidic environments, $Cu^{2+}$

could reduce the fluorescence intensity of C1-C3, and contributions of C4 to total fluorescence intensities gradually increased. For neutral environment, $Cu^{2+}$ exhibited pronounced fluorescence quenching effects on C2, and mild reduction tendencies on C1 and C3. Because of the minimal quenching effects of $Mn^{2+}$, $Ni^{2+}$, and $Zn^{2+}$ on C1-C4, most of the correlation coefficients ($R^2$) between $F/F_0$ and metal ions concentrations were small, indicating the insignificant effects of $Mn^{2+}$, $Ni^{2+}$, and $Zn^{2+}$ on HULIS.

**Table 3 The PARAFAC decomposed fractions of HULIS for winter and summer under three acidity conditions.**

|  |  | C1 | C2 | C3 | C4 | Total fluorescence intensity |
|---|---|---|---|---|---|---|
|  | Acidic | 42% | 19% | 24% | 15% | 805.52 |
| Winter | Weakly acidic | 29% | 24% | 27% | 19% | 869.69 |
|  | Neutral | 22% | 27% | 29% | 22% | 587.49 |
|  | Acidic | 50% | 28% | 20% | 2% | 192.09 |
| Summer | Weakly acidic | 23% | 26% | 18% | 32% | 179.17 |
|  | Neutral | 16% | 32% | 16% | 36% | 161.00 |





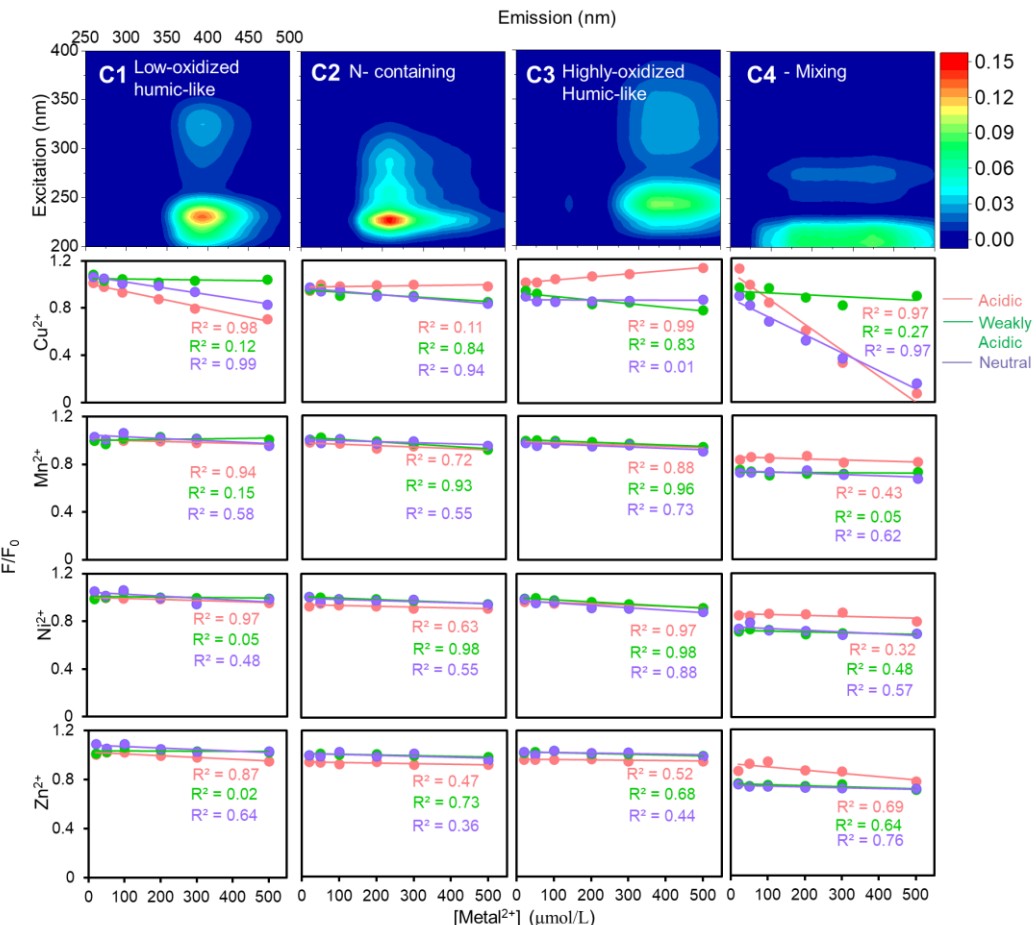

**Figure 6 The changing patterns of four fluorophores with TMs under different acidity conditions for winter HULIS.**

In light of the above analysis, it was obvious that $Cu^{2+}$ demonstrated much more significant effects on optical properties of HULIS than the other three metal ions, which might be because of their disparate complexing abilities (Fan et al., 2021). Li

et al., (2022) found that with increasing $Fe^{3+}$, $Al^{3+}$, and $Cu^{2+}$ concentrations, the fluorescence intensities of highly oxidized HULIS decreased, and those of low oxidized HULIS gradually increased, and they ascribed these phenomena to the fluorophore blue-shifts of highly oxidized HULIS induced by complexation. Likewise, some other studies observed strong bondage between $Cu^{2+}$ and atmospheric organic matter (Wang et al., 2021; Ma et al., 2022). However, the detailed mechanisms causing the differed complexation results of TMs remained unclear.

It is noticed that studies on the effects of TMs on optical properties of fulvic acid and humic acid had been conducted in earlier years in surface water than atmospheric environment, which provided an inspiration to the present study. For example, it was reported in 1980s that $Cu^{2+}$ posed greater fluorescence quenching effects on fulvic-acid than $Co^{2+}$ and $Mn^{2+}$, and it was thus proposed that $Cu^{2+}$ bound internally to fulvic-acid, while $Co^{2+}$ and $Mn^{2+}$ might only externally attached to organics via electrostatic attraction (Ryan et al., 1983). Other studies found that $Cu^{2+}$ had stronger complexation capabilities





compared to the other divalent transition metal ions, such as $Pb^{2+}$, $Cd^{2+}$, $Ni^{2+}$, and $Zn^{2+}$ (Yamashita and Jaffe 2008; Chen et al., 2015; Huang et al., 2018). With application of two-dimensional infrared correlation spectroscopy and synchronous/asynchronous fluorescence spectroscopy analyze, carboxyl and polysaccharide groups were found to bind fast with $Cu^{2+}$; and bondage of $Cu^{2+}$ with phenolic and aromatic carboxyl groups altered the molecule vibration of organics, inducing fluorescence quenching. Besides, $Cu^{2+}$ could bind with several amide and aliphatic groups (protein-like), leading to

fluorescence quenching. More recently, Zhu et al. (2021) suggested that the small atomic radius and paramagnetic properties might result in the unique response of $Cu^{2+}$ to carboxyl groups, thus a strong complexation ability with DOM. Considering the smaller molecules and aromatic structures of HULIS compared to humic substances in water, $Cu^{2+}$ continued its specialty of significant fluorescence quenching effects on HULIS, whereas bondages between $Zn^{2+}$ (or $Mn^{2+}$ and $Ni^{2+}$) and responsive HULIS might be too weak to be observed.

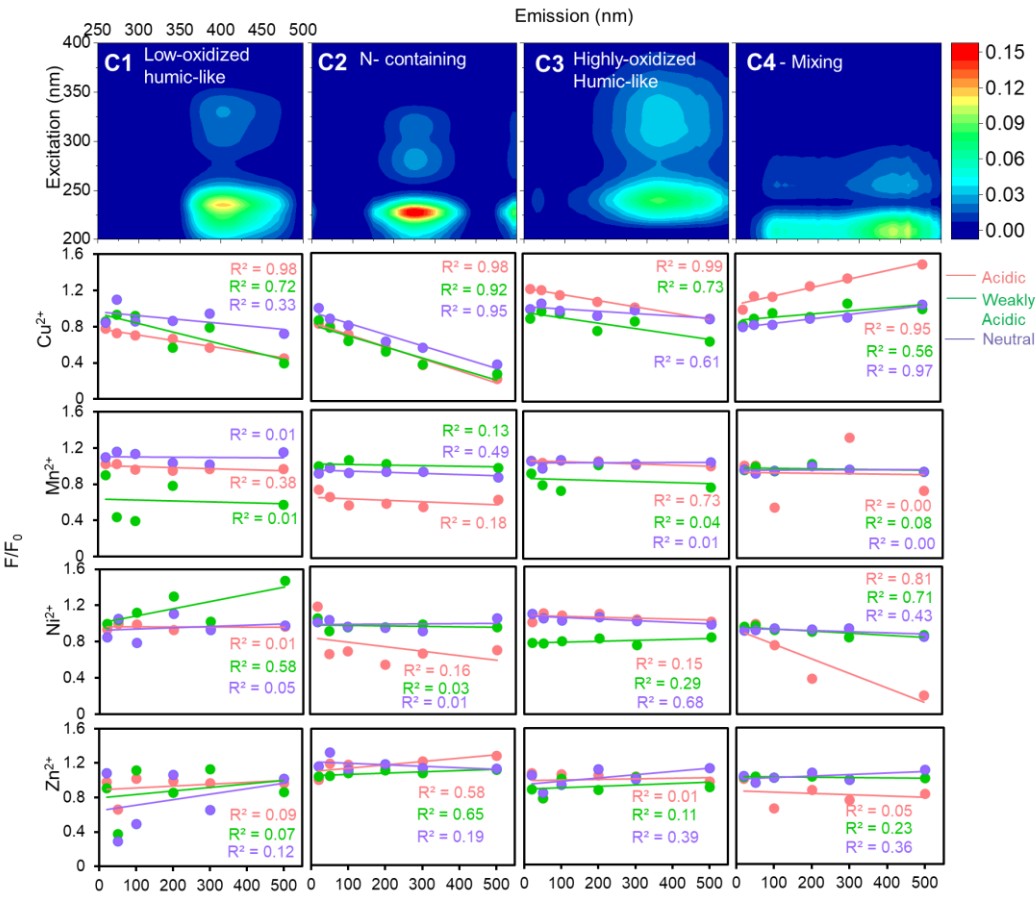


**Figure 7 The changing patterns of four fluorophores with TMs under different acidity conditions for summer HULIS.**



### 3.5 The hidden mechanism of HULIS with addition of TMs - taking Cu2+ as an example

To further understand the complexation mechanisms of $Cu^{2+}$ and HULIS, the structural variations of HULIS before and after adding 100 μM $Cu^{2+}$ were evaluated by $^1$H-NMR and FT-IR methods, respectively (Figure 8 and Figure 9). The $^1$H-NMR results showed that HULIS were rich with protons of H-C, H-C=C-, Ar-H, and H-C-O and functional groups of -OH or -NH, Ar-C=O, Ar-C-O-C, and C-O in both seasons. The addition of $Cu^{2+}$ mainly affected the protons bound with aromatics and oxygen in HULIS, with the signals located between δ7-δ8 and δ3.4-δ4 ppm trailing off, for which the Ar-H signal decreased by 8.9% and 2.1%, and H-C-O signal decreased by 1.4% and 0.8% in winter and summer, respectively. According to the basic principles of NMR spectroscopy, coordination or chemical exchange reactions between HULIS molecules and $Cu^{2+}$ might cause the chemical shift signals vanished or observed as new signals (Günther 2013). Meanwhile, the functional groups of HULIS, revealed by FT-IR spectra, mainly included -OH/-NH, -CH, Ph-C=O, C=O, Ph-C-O-C, and -C-O in both seasons, and the summer HULIS contained more oxygen-containing functional groups than winter HULIS. With the addition of $Cu^{2+}$, the whole FTIR spectra of winter and summer HULIS blue shifted to higher energy, especially for O-containing aromatic groups, but the peak shapes remained unchanged, indicating that $Cu^{2+}$ might not directly lead to the transformation of functional groups, but could result in an enhancement of chemical bonds in HULIS molecules.



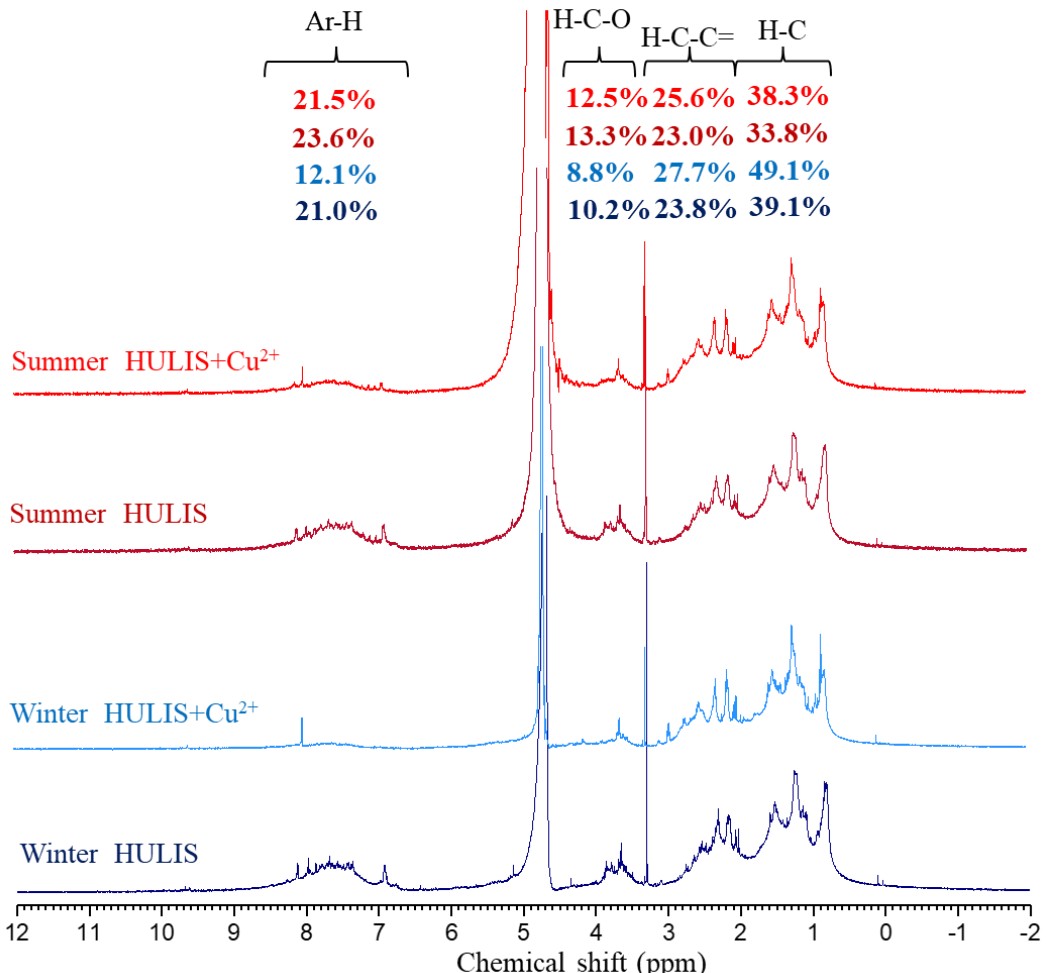

**Figure 8 The $^1$H-NMR spectra of HULIS and HULIS with addition of 100 µM Cu$^{2+}$ in winter and summer.**




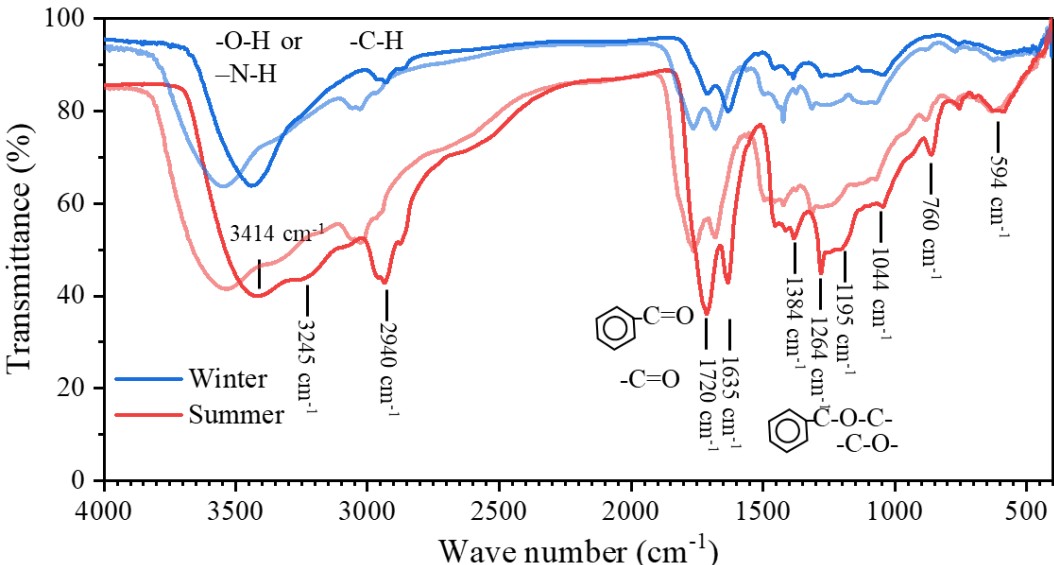

**Figure 9 The FT-IR spectra of HULIS and HULIS with addition of 100 µM Cu$^{2+}$ (light colors) in winter and summer.**

295 Our previous research had characterized the p$K_a$ and corresponding acidic/basic functional groups of winter and summer HULIS, and proposed a hypothetical HULIS structure of unsaturated aromatic main body with pH responsive functional groups (Qin et al., 2022). Briefly, the p$K_a$ of 3.7 – 3.8 and 8.0 – 8.1 were assigned as electron-donating groups (represented by H$_1$A$_1$ and H$_3$A$_3$, respectively), and the p$K_a$ of 5.0 – 6.3 were assigned as electron-withdrawing groups (represented by H$_2$A$_2$) for winter HULIS. Likewise, the p$K_a$ of 6.4 was assigned as electron-donating groups (represented by H$_4$A$_4$) and p$K_a$

300 of 8.4 – 8.5 was assigned as electron-withdrawing groups (represented by H$_5$A$_5$) for summer HULIS. By synthetically considering the versatile optical properties of HULIS with addition of Cu$^{2+}$ under three acidity conditions, and its inherent acid/base groups, the Cu$^{2+}$ and HULIS complexation or interaction mechanisms were explored and the schematic diagram is depicted in Figure 10.

 (1) Firstly, for winter HULIS under acidic environment, the acidity-sensitive HULIS molecules were hard to deprotonate

305 when H$^+$ was crowding around. Most of HULIS existed as forms of H$_1$A$_1$, H$_2$A$_2$, and H$_3$A$_3$. Cu$^{2+}$ could easily be absorbed by the electron-donating groups of HULIS to form Cu-A$_1$H$_1$ complex, or replace the H atom to form Cu-A$_1$, causing static fluorescence quenching of C1 and C4 which slightly increased the lifetime of HULIS. Simultaneously, Cu could alter the charge densities of the fluorescent aromatic molecules, leading to an enlargement of conjugated structures and enhancing the fluorescence of C3.

310 (2) Secondly, for winter HULIS under weakly acidic environment, some of the acidity-sensitive HULIS deprotonated and the solution contained A$_1$$^-$, H$_2$A$_2$, and H$_3$A$_3$. When added with Cu$^{2+}$, A$_1$$^-$ complexed with Cu$^{2+}$ to form Cu-A$_1$, whereas the sensitive structures of H$_2$A$_2$ were electron-donating groups which could hardly provide electrons to Cu$^{2+}$. However, Cu$^{2+}$ could still be attracted by protons in the unsaturated aromatic structures of HULIS, causing dynamic fluorescence quenching of C2 and C3. With regard to the fast-decreasing lifetime and steady AvgSS of HULIS, chemical reactions might not be



involved in these processes, and thus flexible structures of Cu-$A_2H_2$ by electrostatic adsorption or colliding induced energy transfer could be the possible explanation.

(3) Thirdly, for winter HULIS under neutral environment, the acidity-sensitive HULIS existed as $A_1^-$, $A_2^-$, and $H_3A_3$. The deprotonated HULIS had become non-fluorescent, and collision of $A_1^-$ and $A_2^-$ with added $Cu^{2+}$ might not be observed. The fluorescent $H_3A_3$ could provide electrons to $Cu^{2+}$, leading to partially fluorescence quenching effects on C1 and C2, and the

potential complexation mechanism might be similar to $H_1A_1$.

(4) Lastly, for summer HULIS, the quenching mechanism of $Cu^{2+}$ and acidity-sensitive HULIS were similar to those of winter HULIS. Under acidic and weakly acidic environment, $H_4A_4$ and $H_5A_5$ of summer HULIS were not deprotonated, the quenching effects could be resulted from the replacement of H atom in $H_4A_4$ (electron-donating) with $Cu^{2+}$, causing decrease of C1-C3. Under neutral environment, $Cu^{2+}$ complexed with deprotonated $A_4^-$ and aromatic structure of $H_5A_5$ (electron-

withdrawing), mainly causing decrease of C2.

Thus, it was speculated that HULIS with electron-donating groups mainly exhibit fluorescence signals in C1 and C3, which were mainly low and highly oxidized humic-like substances, respectively. Whereas for HULIS with electron-withdraw groups, the fluorescence signals were mainly exhibited in C2, which was N-containing species or protein-like species.

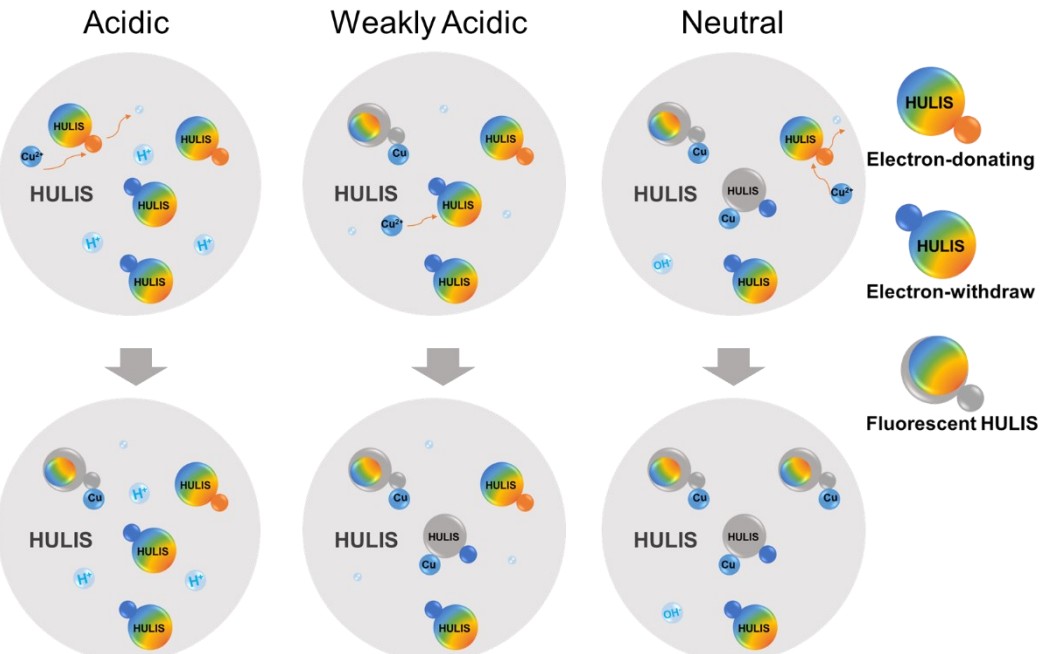

**Figure 10 The schematic diagram of HULIS complexation with $Cu^{2+}$ under different acidity conditions**

**Conclusions and Implications**

In this study, the interrelations between four TMs and HULIS were analyzed by optical methods under acidic, weakly acidic, and near neutral environments. The results elucidated that the light absorption and fluorescence properties of HULIS were




substantially influenced by the solution acidity. TMs either enhance or attenuate the signal, contingent on the metal species,
but with negligible impact on the spectral indices. $Cu^{2+}$ demonstrated the most conspicuous influence on the optical
properties of HULIS, while $Mn^{2+}$, $Ni^{2+}$, and $Zn^{2+}$ proved to be less responsive. The $^1$H-NMR and FT-IR spectra of HULIS
and HULIS-Cu indicated that $Cu^{2+}$ mainly bound with aromatic HULIS, resulting in more compacted molecule structures of
HULIS. The distinct tendencies of PARAFAC-separated fluorophores with increasing $Cu^{2+}$ under three acidity conditions
indicated that electron-donating groups might correspond to low and highly oxidized humic-like substances, and $Cu^{2+}$
coupled with HULIS by substituting H atom, while electron-withdraw groups were N-containing species or protein-like
species, with their fluorescence quenching likely caused by electrostatic adsorption or colliding induced energy transfer.
Results from this study not only revealed that the complexations of HULIS and TMs could be fragile, but also connected the
fluorescence spectra of HULIS with structural characterizations through the micro-effects of TMs on HULIS. HULIS and
TMs are both well-known ROS generative species, and the combined effects of HULIS and TMs on ROS generation could
be synergistic or antagonistic depending on the TMs species (Yu et al., 2018; Lin and Yu 2020). The physical and chemical
interaction between TMs and HULIS might prohibit ROS generation because of their competitive reactions on ROS
generation, but the ROS generation abilities of newly formed TM-HULIS groups could be either accelerated or reduced
depending on the HULIS structures, which are difficult to be distinguished by ROS detection methods. Our results could
provide some perception from chemical analysis perspective on this important issue.

**Code availability**

**Data availability**

The data used in this study are available on the Zenodo data repository platform: https://doi.org/10.5281/zenodo.10460562

**Author contribution**

TJ and WX designed the experiments, QJ and QY carried them out, SZ and GY helped with the experimental issues, LJ and
SS finished the data analysis. QJ prepared the manuscript with contributions from all co-authors, ZL, TJ and WX supervised
the writing, and QT provided professional advices on the discussion section.

**Competing interests**

At least one of the (co-)authors is a member of the editorial board of Atmospheric Chemistry and Physics.



**Acknowledgements**

This work was supported by the National Key Research and Development Program of China (2022YFC370300, 2023YFE0102400, 2022YFC3703402, and 2022YFC3701103), and the Provincial Natural Science Foundation of Hunan (2023JJ30004).

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
