# Peer review of "Measurement Report: Effects of transition metal ions on the optical properties of humic-like substances revealing structural preference"

_EGUsphere, 2023_

## Referee Comment (RC2)

Manuscript Title: **Effects of transition metal ions on the optical properties of humic-like substances revealing structural preference**

Manuscript ID: **egusphere-2023-2632**

**Review Comments:** This manuscript gives insights into light absorbing properties of HULIS with different transition metals at various acidity. The manuscript provides better understanding on light absorption properties of aerosols. The manuscript will be acceptable after addressing below comments. Below provided here some suggestions that improve the readability and presentation of scientific content in the manuscript.

1. Why only four transition metals were considered for the study? Since the objective is to have better understanding of atmospheric processes of transition metals with HULIS. So, why $Fe^{+3}$ was not considered in this study? As $Fe^{+3}$ is involved in the Fenton's reaction and it is one of major reaction known to take place in atmosphere. Then for health risk, Why $Cr^{+6}$ was not considered?

2. The authors give details of the sample collected. Since it is written that samples were collected in winter and summer month. But, did the sample collected covers the whole variation of the seasons or this is only few days sample collection? Please provide details.

3. Firstly, what is the HULIS spectral reproducibility? Add few sentences on uncertainty in the measurements.

4. In the current research, 200-500 nm used to calculate AAE values. Will it make a difference if you choose a high wavelength range to calculate AAE vales? Different groups used different wavelength ranges used to calculate AAE values. Please add the wavelength range used in different studies in the main manuscript.

5. The authors have concluded that $Cu^{2+}$ will have strong influence on HULIS optical properties. But they have not exactly concluded the light absorption properties of HULIS in presence of $Cu^{2+}$ under different acidic conditions. Please elaborate on this.

---

## Author Comment (AC1)

Dear reviewer:

Thank you very much for your recommendation of our manuscript "Measurement Report: Effects of transition metal ions on the optical properties of humic-like substances revealing structural preference". It is very kind of you to give comprehensive and thoughtful advises on the present research. We have carefully addressed all of the comments and explained them as following paragraphs. We have modified the language though out the article as well. Thank you again.

**For Specific Comments:**

**1. Lines 20-21:** Why is the fourth component named as "mixing residentials" ?
**Thank you for your question.** We have checked the context and renamed the component as "mixing residuals" in the manuscript. They now read as follows.

Lines 19-21: The parallel factor analysis (PARAFAC) results extracted four components of HULIS, including low-oxidized humic-like substances (C1), N-containing compounds (C2), highly-oxidized humic-like substances (C3), and the mixing residuals (C4), from the fluorescence spectra in both winter and summer.

Lines 229-231: they were characterized as low-oxidized humic-like substances (C1), N-containing compounds (C2), highly-oxidized humic-like substances (C3), and the mixing residuals (C4), respectively (Chen et al., 2016).

2. Lines 21-25: The subject's adjective phrase is overly extensive in this sentence, compromising the clarity and conciseness of the sentence.
**Sorry for the unconcise description.** We have rewritten the sentence as follows.

Lines 21-24: The spectral characteristic of HULIS with $Cu^{2+}$ additions under three acidity conditions indicated that electron-donating groups of HULIS mainly corresponded to C1 and C3, with $Cu^{2+}$ binding with HULIS by replacing proton, while electron-withdrawing groups of HULIS could correspond to C2, with its connection with $Cu^{2+}$ through electrostatic adsorption or colliding induced energy transfer.

3. Line 31: The transitional word "Besides" seems inappropriate here, because the health effect had already mentioned in the beginning.
**Thank you for your advice.** The word "Besides" have been deleted in the context.

Lines 30-33: HULIS and many TMs, like Fe, Cu, Mn, V, Cr, and Co, have negative impacts on human health since they can promote the generation of reactive oxygen species (ROS), which cause inflammatory response of human respiratory system (Verma et al., 2012; Gali et al., 2015; Lin and Yu 2019).

4. Line 34-26: some references should be cited here to explain the HULIS composition and its environmental effect.

Such as: Seasonal and diurnal variation of PM2.5 HULIS over Xi'an in Northwest China: Optical properties, chemical functional group, and relationship with reactive oxygen species (ROS). Atmospheric Environment, 2022, 268, 118782.

Optical properties, chemical functional group, and oxidative activity of different polarity levels of water-soluble organic matter in PM2.5 from biomass and coal combustion in rural areas in Northwest China. Atmospheric Environment, 2022, 283, 119179.

Optical properties, molecular characterizations, and oxidative potentials of different polarity levels of water-soluble organic matters in winter PM2.5 in six China's megacities. Science of The Total Environment, 2022, 853, 158600.

**Thank you for your advice.** The references are cited in the corresponding places.

Lines 30-35: HULIS and many TMs, like Fe, Cu, Mn, V, Cr, and Co, have negative impacts on human health since they can promote the generation of reactive oxygen species (ROS), which cause inflammatory response of human respiratory system (Verma et al., 2012; Gali et al., 2015; Lin and Yu 2019; Zhang et al., 2022a). HULIS are known as a mixture of macromolecular organic compounds, containing aromatics and aliphatic species with multiple oxygenated functional groups like carbonyls, hydroxyl, nitrate, and nitroxy-organosulfate (Win et al., 2018; Huang et al., 2022; Zhang et al., 2022b). TMs can transfer electrons and participate in chemical reactions or serve as catalyst, especially in atmospheric photochemistry (Mao et al., 2013; Guasco et al., 2014).

5. Lines 45-46: Is it a repeat statement that TMs can affect the light absorption properties of HULIS? when lines 40-42 had firstly emphasized that TMs could mediate the formation of light absorbing SOA.
**Thank you for your question.** We intend to emphasize that the light absorption properties of HULIS may change directly when exposed to TMs solution. We have modified the sentence as follows.

Lines 44-45: Moreover, TMs can also bind with HULIS and directly affect the light absorption properties of HULIS (Fan et al., 2021; Wang et al., 2021).

6. Line 95: How can 5 mg HULIS sample be dissolved into 1 mg D2O solution?
**Sorry for the mistake.** It should be 1 mL $D_2O$ solution here, we have corrected it in the context.

Line 94: About 5 mg HULIS samples were redissolved into 1 mL $D_2O$ for detection.

7. Line 138: An extra line break here.

**Sorry for the mistake.** We have modified the mistake and checked through the whole manuscript.

8. Line 172: The phrase of "the addition of a small amount of any of the four TMs" is excessive redundant.
**Thank you for your advice.** We have shortened the phrase to make it concise.

Line 171: a small amount of any TMs (0–50 μM) could induce evident increase of $MAE_{365}$ at a range of 16% to 20%

9. Lines 178-179: The sentence is hard to read.
**Sorry for the unconcise description.** We have rewritten the sentence as follows.

Lines 178-179: An exception was observed for $Ni^{2+}$ because of its self-absorption between 350–400 nm, leading to the continuously reducing of $AAE_{300-400nm}$ with increasing $[Ni^{2+}]$.

10. Line 185: Ambiguous description of "their complexes don't have structural effects on HULIS.", please clarify it.
**Sorry for the unconcise description.** We have rewritten the sentence as follows.

Lines 183-184: the TMs-HULIS mixtures might not compose complexations or their complexes exhibited no structural difference.

11. Line 231: An extra line break here.
**Sorry for the mistake.** We have modified the mistake and checked through the whole manuscript.

12. Lines 333-334: It is controversial only use "signal" in "TMs either enhance or attenuate the signal", because "spectral indices" could be part of "signal", I believe the author tends to say "signal strength" here.
**Thank you for your advice.** We have corrected the words as suggested.

Lines 330-331: TMs either enhance or attenuate the signal strength, contingent on the metal species, but with negligible impact on the spectral indices.

13. Lines 337-340: The sentence is too long with many comma, it's better to use proper punctuation marks, and make it easier to comprehend.
**Thank you for your advice.** We have tried our best to make the language terse, but it still needs many words to draw a relative precises conclusion, thus, we added ":" to break the sentence, and make it friendly to read.

Lines 334-337: The distinct tendencies of PARAFAC-separated fluorophores with increasing $Cu^{2+}$ under three acidity conditions indicated that electron-donating groups

might correspond to low and highly oxidized humic-like substances, and $Cu^{2+}$ coupled with HULIS by substituting H atom; while electron-withdraw groups were N-containing species or protein-like species, with their fluorescence quenching likely caused by electrostatic adsorption or colliding induced energy transfer.

14. Lines 342-348: In the last paragraph, long descriptions about co-effect of TMs and HULIS on ROS generation, with a short sentence mentioned the implication of this research is somewhat a disappointing end, the authors may add some more specific implications.

**Thank you for your advice.** Because HULIS and TMs are both well-known ROS generative species, we intended to set connections between present research and former research on ROS. We have reorganized the last paragraph as follows.

Lines 338-345: Results from this study not only revealed that the complexations of HULIS and TMs could be fragile, but also connected the fluorescence spectra of HULIS with structural characterizations through the micro-effects of TMs on HULIS. HULIS and TMs are both well-known ROS generative species, and the combined effects of HULIS and TMs on ROS generation can be synergistic or antagonistic depending on the TMs species. The ROS generation abilities of TMs and HULIS are essentially determined by their physical and chemical properties and interactions, the complexation, electrostatic adsorption, and colliding induced energy transfer processes could enhance or prohibit ROS generation and resulted to synergistic or antagonistic effect. Our results could provide some perception from chemical analysis perspective on interactions between HULIS and TMs, however, the exact reaction mechanisms still require further research.

---

## Author Comment (AC2)

Dear reviewer:

We are very grateful for your careful review and insightful comments and suggestions, which have greatly helped to improve the quality and readability of our manuscript. We have addressed all of your concerns and made specific changes to the manuscript as follows.

1. Why only four transition metals were considered for the study? Since the objective is to have better understanding of atmospheric processes of transition metals with HULIS. So, why $Fe^{+3}$ was not considered in this study? As $Fe^{+3}$ is involved in the Fenton's reaction and it is one of major reaction known to take place in atmosphere. Then for health risk, Why $Cr^{+6}$ was not considered?

**Thank you very much for your insightful comments and questions.** We agree that it would be ideal if including some more additional heavy metal species, but limited by the sample amount (it requires a lot $PM_{2.5}$ filter samples to extract HULIS), only four transition metals were selected.

$Cu^{2+}$, $Zn^{2+}$, $Mn^{2+}$, and $Ni^{2+}$ were chosen because of their prevalence in atmospheric environment or significant health risks. $Fe^{+3}$ has been reported to be ROS-generative and photoactive, and the $Fe^{3+}$ chemistry is complex and important that deserves to carry out a separate experiment, thus it is considered in our future research.

$Cr^{6+}$ was not considered because our previous research and recent research on the pollution profile of trace metals in Beijing $PM_{2.5}$ found that Zn, Mn, and Cu were relatively abundant, and Ni posed the highest health risk, so that we choose them as representative metal ions (Zhou et al., 2017; Lei et al., 2021, Hua et al., 2024).

References:
Zhou X, Zeng N, Li Y et al. Chemical Characteristics and Sources of Heavy Metals in Fine Particles in Beijing in 2011-2012. Environmental Science (China). 2017, 38: 4054-4060.
Hua C, Ma W, Zheng F, et al. Health risks and sources of trace elements and black carbon in PM2.5 from 2019 to 2021 in Beijing. Journal of Environmental Sciences, 2024, 142: 69-82.
Lei W, Zhang L, Xu J, et al. Spatiotemporal variations and source apportionment of metals in atmospheric particulate matter in Beijing and its surrounding areas. Atmospheric Pollution Research, 2021, 12: 101213.

2. The authors give details of the sample collected. Since it is written that samples were collected in winter and summer month. But, did the sample collected covers the whole variation of the seasons or this is only few days sample collection? Please provide details.
**Thank you very much for your question.** The $PM_{2.5}$ samples were collected over a one-month period each during the winter and summer seasons, and HULIS were subsequently isolated from a combination of 30 mixed samples. We have added some additional information in section 2.1 to provide a more detailed description, they are

shown as follows.

Lines 57-59: In the present research, two concentrated HULIS solutions were extracted from $PM_{2.5}$ samples collected in a winter and summer month **(30 samples in each season)** in Beijing, and the detailed sample information can be found elsewhere (Qin et al., 2022).

3. Firstly, what is the HULIS spectral reproducibility? Add few sentences on uncertainty in the measurements.
**Thank you very much for your question.** The UV-Vis and fluorescence measurements in this research exhibit excellent spectral reproducibility and low uncertainties. We have added details regarding these aspects in the Methods section. They are read as follows in the manuscript.

Lines 90-92: The limit of detection for UV-Vis and fluorescence analysis were estimated as 2 times of their respective standard deviation of blanks, which were 0.005 and 0.01 absorption unit, respectively.

4. In the current research, 200-500 nm used to calculate AAE values. Will it make a difference if you choose a high wavelength range to calculate AAE vales? Different groups used different wavelength ranges used to calculate AAE values. Please add the wavelength range used in different studies in the main manuscript.
**Thank you very much for your suggestions.** Sorry that some of our descriptions are not prominent enough in the manuscript. We used the wavelength range of 300-400 nm to calculate $AAE_{300-400}$, which were commonly used range in previous research, the calculation was in section 2.3.1 (Lines 102-106). We also added some relating references in the context. They are now read as follows.

Line 104-108:
$$MAE = (A_\lambda - A_{700}) \times \frac{\ln(10)}{C_{spcies} \times L} \qquad (1)$$
$$A_\lambda = K\lambda^{-AAE} \qquad (2)$$
Where $A_\lambda$ is light absorbance at wavelength λ, $C_{species}$ is the chemical concentration of organic compounds (WSOC and HULIS in the present research), L is the light path length (1 cm), $K$ is a scaling constant, and the fitting wavelength of AAE is 300–400 nm.

Lines 141-146: The MAE and AAE of HULIS under three acidity levels in both seasons were consistent with those reported in our previous research, with the average $MAE_{365}$ at 0.011±0.00 in winter and 0.005±0.001 in summer, and the corresponding average $AAE_{300-400nm}$ at 6.46±0.86 and 6.97± 0.83, respectively (Qin et al., 2022), which were comparable with those of Gosan Korea (Kirillova et al., 2014), Hongkong China (Ma et al., 2019), and Tibet China (Wu et al., 2019), but lower than those of biomass burning sources (Park et al., 2016).

5. The authors have concluded that Cu2+ will have strong influence on HULIS optical properties. But they have not exactly concluded the light absorption properties of HULIS in presence of Cu2+ under different acidic conditions. Please elaborate on this.

**Thank you very much for your suggestion.** We have carefully reviewed the conclusion section of our manuscript and agree with your assessment that it was lacking information about the effect of $Cu^{2+}$ under different pH conditions. We have now added the following text to the conclusion.

Lines 342-343: $Cu^{2+}$ could promote light absorption ability of HULIS at weakly acidic to neutral environment, but the effect was negligible under acidic environment.

---

## Author Response (AR2)

Dear reviewer:

Thank again for your comprehensive review and thoughtful advises on the present research. We have carefully addressed all of the comments and explained them as following paragraphs, and hoping that the new version of this manuscript will meet your satisfaction.

1. Since this study has only focused on the HULIS and optical properties measurements in China, thus the title should state that the study is for China. Otherwise, the authors should also consider other studies on HULIS or optical properties around the globe or at least in Asian Countries.

**Thank you for your advice.** As our samples were only collected in Beijing, we have amended the title to clarify that this is a case study conducted in Beijing. The revised manuscript title reads as follows.

**Lines 1-3: "Measurement Report: Effects of transition metal ions on the optical properties of humic-like substances (HULIS) revealing structural preference-A case study of $PM_{2.5}$ in Beijing, China".**

2. I do not find authors response to first question i.e "Why only four transition metals were considered for the study? Since the objective is to have better understanding of atmospheric processes of transition metals with HULIS. So, why Fe+3 was not considered in this study? As Fe+3 is involved in the Fenton's reaction and it is one of major reaction known to take place in atmosphere. Then for health risk, Why Cr+6 was not considered?" satisfactory. They should provide proper reasoning in the response as well as in the manuscript.

**Thank you for your advice.** Thank you for your feedback. We apologize for the previous unsatisfactory response. Our current research primarily focuses on the effects of metal ions on the optical properties of HULIS. Therefore, we selected $Cu^{2+}$, $Zn^{2+}$, $Mn^{2+}$, and $Ni^{2+}$ due to their abundance in Beijing $PM_{2.5}$ and their previously confirmed quenching effects on water-soluble organic compounds in our prior studies. Upon reflection, we acknowledge that our previous description, stating "Four transition metal ions including $Cu^{2+}$, $Zn^{2+}$, $Mn^{2+}$, and $Ni^{2+}$ were chosen as representative metal ions that might be reactive to HULIS, considering their richness in the atmospheric environment or significant health risks (Fan et al., 2021; Wang et al., 2021)," was inaccurate. Indeed, the significance of these four metals is lower than that of $Fe^{3+}/Fe^{2+}$ and $Cr^{6+}$ concerning both their abundance in $PM_{2.5}$ and associated health risks. Therefore, we have revised this sentence in the manuscript.

**Lines 73-74:** "Four transition metals ions that are relatively rich in Beijing $PM_{2.5}$ and can complex with HULIS were considered including $Cu^{2+}$, $Zn^{2+}$, $Mn^{2+}$, and $Ni^{2+}$ (Wang et al., 2021)."

While $Fe^{3+}$ and $Cr^{6+}$ were not considered in our current study, their importance in atmospheric research cannot be overlooked. Therefore, we have also emphasized their significance in the "Conclusions and Implications" section at the end of the manuscript.

**Lines 361-364:** "Our results provide perceptions from chemical analysis perspective on interactions between HULIS and TMs, however, due to the intricate nature of $PM_{2.5}$, many crucial transition metals, such as $Fe^{2+}/Fe^{3+}$ and $Cr^{6+}$, which play pivotal roles in the chemical transformation of $PM_{2.5}$ or pose high risks to human health, are not addressed in the current study. Furthermore, comprehensive research on worldwide HULIS samples is also imperative for future investigations."

3. They have said that they have done one month of sampling during both the winter and summer season. So, they should also mention this in the manuscript on line 57-59.

**Thank you for your advice.** We included "(30 samples in each season)" in line 58 of the previous version of the manuscript. However, it appears that this addition may still lead to confusion. Therefore, we have revised this sentence again as follows.

**Lines 57-59:** "In the present research, two concentrated HULIS solutions were extracted from daily $PM_{2.5}$ samples collected in summer and winter of 2016 (30 samples in each season) in Beijing, and the detailed sample information can be found elsewhere (Qin et al., 2022)."

4. The quality of the manuscript can be further improved by incorporating the comparison with previous studies done around the globe.

**Thank you for your advice.** We have added some comparison in the "Results and Discussion" section as follows.

**Lines 174-178:** "For winter HULIS, under acidic environment, the addition of $Mn^{2+}$, $Ni^{2+}$, and $Zn^{2+}$ at low concentrations (< 50 μM) could induce a decrease of 14%–16% in $MAE_{365}$, and further increasing TM concentrations had little effects on $MAE_{365}$, whereas $Cu^{2+}$ showed minimal effects on $MAE_{365}$, which only decreased $MAE_{365}$ by 5% with increasing TM concentrations; **these results were consisted with former study who found that the light absorption capacity of the atmospheric HULIS solution in the $Cu^{2+}/Al^{3+}/Zn^{2+}$-coupled system can be enhanced to 5 − 35% (mean: 19%) (Li et al., 2022).**"

**Lines 283-286:** "The [1]H-NMR results showed that HULIS were rich with protons of H-C, H-C=C-, Ar-H, and H-C-O and functional groups of -OH or -NH, Ar-C=O, Ar-C-O-C, and C-O in both seasons, **which were similar to the results of former researches (Fan et al., 2013; Hawkins et al., 2016; Kumar et al., 2017).**"

**Lines 290-293:** "Meanwhile, the functional groups of HULIS, revealed by FT-IR spectra, mainly included -OH/-NH, -CH, Ph-C=O, C=O, Ph-C-O-C, and -C-O in both seasons, and the summer HULIS contained more oxygen-containing functional groups than winter HULIS **because of the high atmospheric oxidization capacity (Chen, et al. 2016; Haynes et al., 2019).**"

**Lines 335-339:** "Besides, although very few studies can explicit the interaction mechanisms between HULIS and heavy metals, the hypotheses of present study are aligned with a recent soil study, exploring the retention of heavy metals by humic acid using atomic force microscopy who found that some metals ($Pb^{2+}$ and $Cd^{2+}$) formed strong adhesion stems from the synergistic metal-humic acid complexation and cation-π interaction at pH 5.8, leading to significant retention, while other metals ($As^{5+}$ and $Cr^{6+}$) only bound weakly through hydrogen bonds (Wang et al., 2024)"